# Developing an intervention to increase REferral and uptake TO pulmonary REhabilitation in primary care in patients with chronic obstructive pulmonary disease (the REsTORE study): mixed methods study protocol

Frances Early,[1] Patricia Wilson,[2] Christi Deaton,[3] Ian Wellwood,[3] Terry Dickerson,[4] James Ward,[4] Lianne Jongepier,[5] Ruth Barlow,[6] Sally J Singh,[7] John Benson,[3] James Brimicombe,[3] Lois Kim,[3] Hena Haque,[1] Jonathan Fuld[1]

For numbered affiliations see end of article.

**Correspondence to**
Frances Early;
frances.early@addenbrookes.nhs.uk

## ABSTRACT

**Introduction** Chronic obstructive pulmonary disease (COPD) is a progressive lung disease associated with breathlessness, inability to exercise, frequent infections, hospitalisation and reduced quality of life. Pulmonary rehabilitation (PR), providing supervised exercise and education, is an effective and cost-effective treatment for COPD but is significantly underused. Interventions to improve referral and uptake have been tested and some positive results reported. However, interventions are diverse and no clear recommendations for practice can be made. This study aims to understand the challenges to referral and uptake in primary care, where most referrals originate, and to develop a flexible toolkit of resources to support referral and uptake to PR in primary care in the UK.

**Methods and analysis** This is a mixed methods study informed by normalisation process theory and burden of treatment theory. In the first phase, general practitioners, practice nurses and PR providers will be invited to complete an online survey to inform a broad exploration of the topic areas. In phase 2 interviews and focus groups will be conducted with patients, healthcare professionals (HCP) in primary care, PR providers and commissioners to gain an in-depth understanding of the issues and needs. Toolkit development in phase 3 will draw together the learning from phases 1 and 2 and employ an iterative development process to build the toolkit jointly with patients and HCPs. It will be tested in primary care for usability and acceptability.

**Ethics and dissemination** The study has ethical and Health Research Authority approval (Research Ethics Committee reference number 17/EE/0136). It is registered with the International Standard Registered Clinical/Social Study Number (ISRCTN) registry (trial ID: ISRCTN20669629, assignment date 20 March 2018, trial start date 1 April 2016). Dissemination will be aimed at patients, carers/families, service providers, commissioners and national interest groups. Methods will include conferences, presentations, academic publications and plain English reports and will be supported by the British Lung Foundation.

**Trial registration number** ISRCTN20669629 ; Pre-results.

## Strengths and limitations of this study

► The study explicitly uses normalisation process theory to inform data collection methods, data analysis and toolkit development which will maximise potential for the toolkit to be successfully implemented in practice.

► Patients who are unable to communicate in English, Hindi or Urdu will be excluded from the study.

► Patients who have declined pulmonary rehabilitation may be challenging to engage and so we will work closely with primary care to recruit this group.

► An iterative design process enables rapid development of the toolkit in a way that readily allows it to be refined and tailored to the needs of users and the overall system, resulting in a design of higher quality which is more likely to stand the test of time.

► Outputs from survey and qualitative data will support identification of design priorities at the start of development which will provide focus for the iterative process.

## INTRODUCTION

Chronic obstructive pulmonary disease (COPD) is a progressive lung disease associated with breathlessness, inability to exercise, frequent infections and hospitalisation. Approximately 1.2 million people were living with diagnosed COPD in England in 2012[1] and annual direct healthcare costs of COPD in England have been estimated to increase from £1.50 billion in 2011 to £2.32 billion in

2030.[2] Pulmonary rehabilitation (PR), providing supervised exercise and education, is an effective treatment for COPD that leads to clinically significant improvements in exercise capacity, symptoms and health-related quality of life in patients who experience disabling breathlessness.[3] PR also supports self-management skills[4] and results in fewer and shorter hospital admissions[5] and readmissions.[6] It is a cost-effective treatment[7] recommended for patients who are functionally disabled by COPD.[8]

However, despite published guidelines,[8 9] PR is significantly underused. In England and Wales in 2013/2014 the estimated prevalence of patients with COPD eligible for PR was 446 000 but only 68 000 referrals were received by PR programmes. Of those patients referred, 31% did not attend for assessment.[10] There is an urgent need to review PR referral pathways, healthcare professional (HCP) training, information for patients and referrers, and barriers to patient access, particularly in primary care in England and Wales where 51% of referrals originate.[10]

Referral rates are impacted by a range of factors including difficult referral processes, lack of information about PR and unclear roles and responsibilities regarding referral.[11–24] A recent systematic scoping review of barriers and enablers to PR referral identified the most common barriers to be low knowledge of, or disbelief in, the benefits of PR and low knowledge of the referral process; other frequently identified barriers were low knowledge of the eligibility criteria, low awareness of PR and a belief that gaining behaviour change in reluctant patients would be too difficult.[25] The same review reported that the most frequently identified enablers of PR referral were PR training, mentoring or experience in PR; other enablers were PR awareness events, reminders and a streamlined referral process. Uptake by patients is impacted by the quality of the HCP conversation about PR and patients' beliefs about the benefits, as well as timing, location and transport.[26–28]

Interventions to improve referral and uptake have been reported[29–42] and the results of these have been summarised in a systematic review.[43] In this review, 4 out of 10 studies that measured referral reported statistically significant increases in referral: the interventions included a patient-held care quality scorecard,[34] clinician education[35 41] and mandatory monitoring of quality indicators in a hospital setting.[37] Two out of four studies measuring uptake reported statistically significant increases in uptake: a patient manual summarising evidence of COPD treatments[39] and individualised care planning supported by nurses and general practitioners (GP).[40] Most of the studies measured referral or uptake to PR in the context of multifaceted evidence-based management of COPD. Clear recommendations for practice could not be made due to the diverse range of study designs and most study designs carrying a high risk of bias.[43]

This paper presents the protocol for a mixed methods study to support the improvement of referral and uptake to PR from primary care. The study runs from April 2016 to May 2019.

## Aims and objectives

The aim is to understand the needs of HCPs and patients in general practice concerning the challenges to referral and uptake, and then use this understanding to develop a flexible toolkit of resources to support referral and uptake of PR within primary care.

The research questions are:
1. From HCPs and patients' perspectives how can barriers to PR referral and uptake be overcome and facilitators of utilisation be supported?
2. What components do HCPs and patients believe should be included in a toolkit to increase PR referral and uptake?
3. How acceptable and useable is the toolkit for HCPs and patients as a means to increase referral and uptake of PR?

The objectives to address the research questions are:
A. Conduct empirical research to identify factors that HCPs and patients believe will overcome barriers and facilitate referral and uptake to PR in primary care.
B. Synthesise evidence from empirical research and our systematic literature review to develop the toolkit collaboratively with patients and providers.
C. Conduct usability and acceptability testing of the toolkit in a sample of primary care practices to assess performance and gather data to provide a basis for future efficacy and economic analysis.

## Guiding theory

The research design is informed by normalisation process theory,[44 45] an established theoretical framework for understanding implementation processes through multiple stakeholder perspectives. Normalisation process theory will provide the theoretical underpinning for data collection, analysis and interpretation, and toolkit development. The four key constructs of normalisation process theory will be applied to the PR context to understand how PR referral becomes embedded in clinical practice:
A. Coherence: how PR is understood by clinicians.
B. Cognitive participation: clinicians' buy-in and support for PR.
C. Collective action: the operational work required for PR referral.
D. Reflexive monitoring: how PR is appraised by clinicians and how work is reconfigured to enable it to happen.

Closely aligned to normalisation process theory, burden of treatment theory will be used to understand patients' experiences.[46] This theory provides a structural model of the relationship between treatment burden and patients'/carers' capacity to undertake work required to manage their condition, and helps to understand variations in healthcare utilisation and adherence.[47] Health service responses to reduce this burden will be understood through the minimally disruptive medicine model, a patient-centred approach in which patients and professionals work together to advance patient goals while imposing the smallest possible treatment burden.[48]

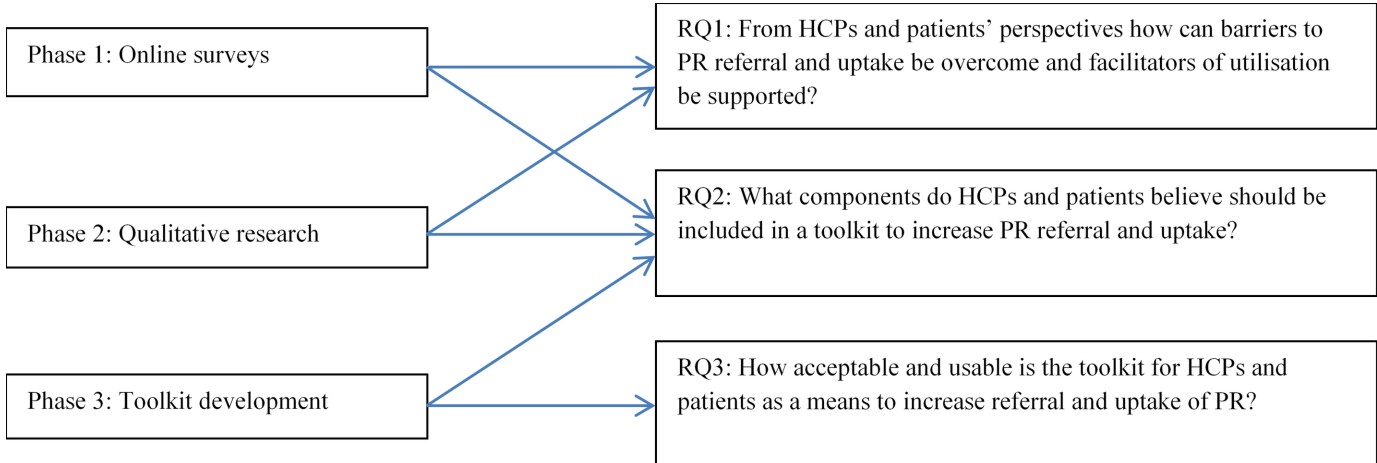

**Figure 1** Mapping of research questions and phases. HCP, healthcare professional; PR, pulmonary rehabilitation.

Key elements of burden of treatment theory and minimally disruptive medicine mirror those of normalisation process theory and will inform understanding of how patients:

A. Make sense of their health conditions, tests and treatments.
B. Enrol support, plan to attend healthcare visits and enact self-care activities.
C. Operationalise the work of attending visits and self-managing their care.
D. Monitor, appraise and evaluate the worth of the work they are doing.

## METHODS AND ANALYSIS
### Study overview

The study comprises three phases:

Phase 1: Online survey of HCPs in general practice and online survey of PR providers.

Phase 2: Qualitative research with patients, HCPs and commissioners.

Phase 3: Toolkit development research.

Figure 1 shows how the phases map to the research questions.

Phases 1 and 2 will address the topic areas listed below to identify facilitators and barriers to referral and uptake and explore current practice and ideas for improvement. The data collection and analysis during these phases will be informed by normalisation process theory in order to explicate how current ways of working, and the impact of barriers and facilitators, are embedded and sustained in practice. The application of normalisation process theory will also inform understanding of how ideas for improvement can be embedded and sustained. We will sample from four participant groups (patients, HCPs, PR providers and commissioners) in order to compare and contrast the perspectives and needs of these groups in relation to PR referral. The topic areas are:

▶ Influences on HCP decisions to refer to PR.
▶ Reasons why patients take up or decline a referral.

▶ Methods to make it easier to refer to PR and take up a referral.
▶ Tools/strategies in use to prompt referral and uptake.
▶ Support or training that would improve counselling of patients regarding PR.
▶ Ways to ensure effective communication between primary care and PR providers regarding referrals.
▶ Priorities and design considerations that the toolkit should address.

Findings from phases 1 and 2 will inform toolkit development in phase 3. The final study output will be a toolkit tested in clinical practice for usability and acceptability which, if validated, will be ready for wider testing.

### Phase 1: online surveys of HCPs in general practice and of PR providers

The surveys enable a broad exploration of the topic areas. Pulmonary disease or long-term condition leads in all GP practices in the East of England (n=approximately 455) will be invited to complete the general practice survey by the Clinical Research Network Eastern. The study team will send invitations to all PR providers (n=21) in the East of England PR Network to complete the PR provider survey.

Survey design is informed by normalisation process theory and includes items adapted from the NoMAD tool.[49] We sought input to the design from practice nurses and PR providers and piloted each survey. Items are dichotomous, multiple response or Likert scales, supplemented by free-text questions (see online supplementary materials 1 and 2). We will invite HCPs to submit relevant resources that they recommend. Surveys will be implemented on the Online Surveys platform[50] and a prize draw of £200 in online shopping vouchers will incentivise returns.

Quantitative data will be uploaded to SPSS (V.25) for descriptive analysis including frequencies and comparison of responses by different healthcare providers and levels of experience. General practice data will be compared by Clinical Commissioning Group area and

mapped onto referral and uptake data from PR services where this is available.

Free-text responses will be imported into NVivo V.12 software. Analysis will follow a 'framework' approach as described by Ritchie *et al*.[51] Constructs from normalisation process theory will inform a deductive coding structure supplemented by inductive coding to allow unexpected findings to emerge. Codes will be grouped into categories and a working analytical framework developed which will be applied to all data. Data will be summarised by category in framework matrices on which the final interpretation will be based.

## Phase 2: qualitative research with patients, HCPs and commissioners

The purpose of the qualitative research is to enable an in-depth exploration of the topic areas with the following groups:

1. Patients who have accepted a PR referral (focus groups).
2. Patients who have declined a PR referral (interviews).
3. Patients who have not been referred to PR (interviews).
4. HCPs in primary care who refer patients to PR (interviews and/or focus groups).
5. Commissioners of PR services (interviews). National Health Service (NHS) commissioners are responsible for planning and purchasing healthcare services for their local populations.
6. Physiotherapists and/or nurses who deliver PR (focus group).

### Inclusion criteria

Patients will be eligible if they are resident in the East of England, have a diagnosis of COPD with stable disease, are eligible for PR as defined by guideline recommendations[8] and able to read and write in English. Patients who have accepted a PR referral will be eligible if they have attended only one programme. In addition, we will conduct interviews and focus groups in two geographical areas specifically targeting patients of South Asian heritage. For these activities, patients will be eligible if they do not have the ability to read or write in English but can communicate, read and write in Hindi or Urdu.

### Sampling

We will begin by identifying four PR services within the East of England based on utilisation of commissioned places in 2014/2015. We will select the two services with the highest utilisation rates and the two with the lowest utilisation rates and then apply convenience sampling to identify participants as follows:

Patients and HCPs: Following convenience sampling within the PR services and the general practices that refer to them, we will recruit patients who have accepted a PR referral, patients who have declined or have not been referred to PR and HCPs in primary care who refer patients to PR. We will aim to achieve a gender balance among patients.

**Table 1** Estimated sample sizes for each group of participants

| Participant group | Estimated sample size |
| --- | --- |
| Patients who have accepted a PR referral (six focus groups of up to eight participants) | Up to 48 |
| Patients who have declined a PR referral (interviews) | Up to 9 |
| Patients who have not been referred to PR (interviews) | Up to 9 |
| HCPs in primary care who refer patients to PR (interviews and/or focus groups) | Up to 34 |
| Commissioners of PR services (interviews) | 4 |
| Physiotherapists and/or nurses who deliver PR (focus group) | Up to 10 |

HCP, healthcare professional; PR, pulmonary rehabilitation.

Commissioners: We will use convenience sampling to identify a commissioner of each of the four PR services.

PR providers: We will use convenience sampling within the East of England PR Network to identify PR providers across the region.

We will also select two geographical areas in the East of England that have populations of South Asian heritage. We will use convenience sampling within the PR services and general practices to identify patients of South Asian heritage who have accepted a PR referral, patients who have declined or have not been referred to PR and HCPs in primary care who refer patients to PR. We will aim for gender balance among patients.

We envisage that it may be challenging to recruit some participants, including patients who have declined PR and clinicians in primary care with busy and demanding schedules, and so convenience sampling will allow us flexibility in achieving the required numbers of participants. To enhance validity, the final sample size will be determined by the need to reach data saturation. In a study of patients referred to PR, Arnold *et al*[52] achieved saturation with 24 patients. Table 1 shows an estimation of the sample sizes to be aimed for. If necessary, more participants will be sought in order to achieve data saturation or the final number may be lower if saturation is achieved with a lower number.

Focus groups have the benefit of drawing out shared experiences among participants and we have planned focus groups where we have a reasonable expectation of being able to convene sufficient numbers of participants. Where we do not expect this to be the case we have planned for interviews.

### Recruitment

Patients who have accepted a PR referral: Each PR service will identify eligible patients, issue the invitation to attend the focus group and confirm attendance.

Patients who have declined a PR referral or who have not been referred to PR: General practices will identify

patients from their COPD register and the invitation will be made by a clinician known to the patient. Telephone interviews, home visits or an interview on general practice premises will be offered to encourage participation. Interested patients will agree to be contacted by the study team who will make arrangements for interview.

HCPs in primary care: Clinical Research Network Eastern will invite HCPs to participate. Interested practices will return an expression of interest form to the Clinical Research Network who will inform the study team. The team will then arrange the interview or focus group.

Commissioners of PR services: The study team will identify and invite commissioners with support from Clinical Research Network Eastern.

Physiotherapists/nurses who deliver PR: The study team will invite participation through the East of England PR network.

### Informed consent

Study team researchers will take written informed consent prior to commencing interviews or focus groups. Where telephone interviews are conducted consent will be taken verbally and recorded on audio prior to the interview. Consent will also be taken from carers, family members or friends who accompany participants as they may provide information about the participant during the course of the interview or focus group. Participants will be free to withdraw from the study at any time.

### Data collection

Question guides for interviews and focus groups (online supplementary materials 3–8) are developed from the relevant topic areas. Patient focus groups will be held at community or healthcare venues and interviews will be at the patient's home, a healthcare venue or conducted by telephone. Focus groups in general practice may be challenging to organise due to competing work schedules and geographical location so we will offer one-to-one interviews as well, on general practice premises or by telephone. The PR provider focus group will be held at a healthcare venue and commissioner interviews at the commissioner's place of work or by telephone. Interviews and focus groups will be audio recorded and transcribed verbatim.

### Data analysis

Transcriptions and field notes will be imported into NVivo V.12 software for data management and analysis. Analysis will follow a 'framework' approach.[51] Normalisation process theory constructs will inform a deductive coding structure supplemented by inductive coding to allow unexpected findings to emerge. Scripts will be read, re-read and coded by two researchers. Codes will be grouped into categories and a working analytical framework developed which will be applied to all transcripts. Data will be summarised by category in framework matrices on which the final interpretation will be based.

### Quality criteria

Structured and standardised question guides will ensure consistency. Transcripts will be coded independently by two researchers. Reviewing and revising the analysis with a patient reference group will provide a credibility and reliability check.

### Phase 3: toolkit development research

The purpose of this phase is to synthesise findings from phases 1 and 2 and incorporate the learning into the toolkit development.

### User group

A user group of up to fifteen HCPs and up to fifteen patients recruited from research participants, volunteers from patient and public involvement (PPI) events and British Lung Foundation networks will work with the study team, a web developer and an expert in design and development from the Engineering Design Centre at the University of Cambridge to develop the toolkit. We will offer a range of ways to participate in testing and feedback, for example, online, face-to-face meetings, visiting members, holding satellite meetings at professional conferences or service/training meetings. Written informed consent will be taken for all participants.

### The design process

A rigorous, iterative design process (Explore, Create, Evaluate)[53] (figure 2) will track the user journey from initial concept to building the toolkit, including the development of criteria for acceptability and usability testing. To explore potential toolkit content and structure the needs, a review will take place of the summaries of themes identified from the survey and qualitative research. This will be underpinned by normalisation process theory and burden of treatment theory constructs, and will be supported further by the outputs from the literature review and work by other expert groups, for example, the British Thoracic Society PR Quality Improvement Group. A consensus method adapted from nominal group technique[54] will be used to consider and vote on priorities. Content and media will be developed during the user-centred design process but we are aiming for a simple and practical toolkit with information about ways to impact referral and uptake. The toolkit is intended for direct use by HCPs in primary care with the aim of enhancing their understanding of PR, providing resources to support a meaningful and motivational conversation about PR with the patient and to facilitate the referral process. Examples could include key messages about the benefits of PR, electronic prompts/templates to prompt referral, links to guidelines and decision support tools and printing options. It is important that we do not pre-empt the design process, in terms of content or medium, as this must be coproduced with users. However, based on what we know from the literature about barriers to referral and uptake we would envisage important features to include: clear information about PR and its benefits and eligibility

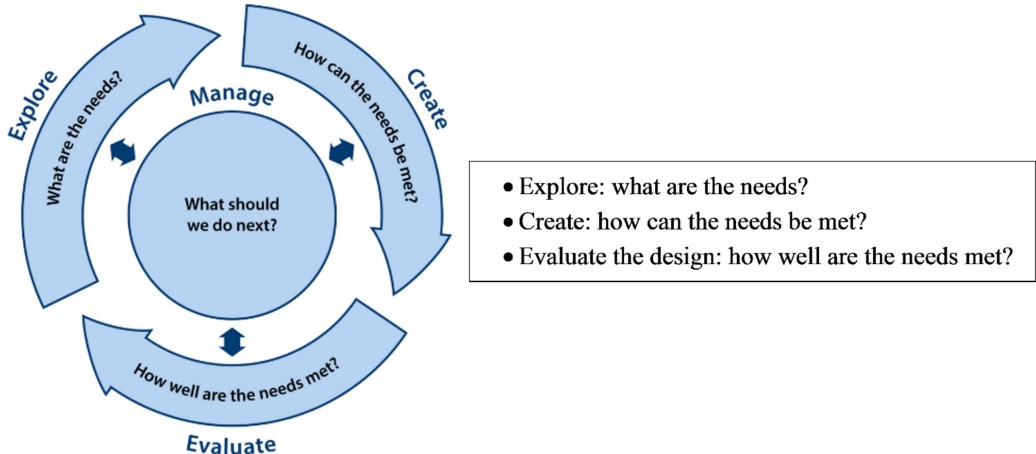

**Figure 2** The fundamental questions of design are solved through successive cycles of exploring needs, creating concepts and evaluating options, guided by project management.[51]

criteria; prompts and tools to facilitate the referral process; tools to support an exploration of the patient's needs and beliefs about PR; tools and information to support the HCP to motivate patients and encourage uptake. The toolkit may also include information for patients that can be shared with them by the HCP. It is not intended that patients will access the toolkit directly. Design will initially be solution neutral and will then be translated into practical solutions. Practical resource constraints will be considered and any user-defined needs that could not be addressed will be recorded. The group will also define initial assessment criteria for usability and acceptability testing.

Agile methods[55] will be used to create the toolkit. The study team will build initial web pages in the live location to develop concepts and stimulate ideas. Early and subsequent releases of the toolkit will be continuously explored, created, evaluated and fed back on with the user group, alongside continuous web development in short time cycles over 6 months. Usability and acceptability assessment criteria will determine the feedback requested and with each round of feedback the criteria will be open to further refinement as users explore and identify their practical needs. The toolkit will be built on a WordPress platform, a free open-source website infrastructure and content management system, and hosted on University of Cambridge web servers by University Information Systems. A web development expert from the University of Cambridge will lead the technical development. It will be designed to interface with working systems (in ways to be identified during the research, eg, linked through SystmOne,[56] integrated into annual review) and a flexible design will enable ongoing development.

### Data collection

During toolkit development subjective and objective data will be gathered from user group participants to assess the user-defined usability and acceptability criteria. The methods of testing acceptability and usability will be those best suited to measure those criteria.

Therefore, at this stage, we cannot predetermine what those methods will be. However, we envisage considering a range of approaches including retrospective methods, for example, written feedback and interviews, and real-time methods, for example, thinking aloud techniques and objective methods, for example, user observation. Testing will take place after each iteration of the design process.

Following development we will work with five general practices to identify the HCP most involved in PR referrals and train them to use the toolkit for 4 weeks in a clinical setting. Feedback will be sought relating to the user-defined assessment criteria plus user experience, perceived understanding of PR and referral criteria, technical issues and nurse time to implement the toolkit. Each HCP will complete a short questionnaire following use of the toolkit with patients and an interview at the end of the 4-week period. Patients with whom the HCP has used the toolkit will complete a short evaluation questionnaire to assess the impact on the patient experience (online supplementary material 9). This will be returned directly to the study team with a stamped addressed envelope. Questionnaires will reflect the evaluation criteria. Written informed consent will be taken from HCPs and patients. During this testing period, we will collect usage metrics from the system and data on referral numbers and uptake compared with baseline.

### Data analysis

Quantitative data will be uploaded into Excel and descriptive analysis will be conducted to identify frequencies. Interviews will be audio recorded, transcribed and uploaded to NVivo V.12 software for data management and analysis. Data analysis will follow procedures as described in phase 2 with the addition that the user-defined usability and acceptability criteria, together with normalisation process theory constructs, will inform the development of a deductive coding structure.

## Patient and public involvement

We involved a range of people in PPI activities to inform the study design and plain English communications. Forty-seven people from PR programmes and patient support groups attended across six meetings, we consulted four members of the Cambridge University Hospitals PPI panel and noted feedback from patient participants in meetings about the future of services throughout the Cambridgeshire and Peterborough healthcare system. A need for increased access to PR and the value of locating the study in primary care were seen as important. As a result we engaged a GP coapplicant to the study. PPI input is ongoing and we have established a PPI group, supported by a member of the study team, who will review research materials, feed back on the research analysis and support dissemination. We offer travel expenses to PPI meetings. The British Lung Foundation will also support active engagement of their patient members.

We also sought feedback on the research priorities from practice nurses at professional development meetings. They wanted easy access to good quality information and methods to engage patients, and recommended an emphasis on patient benefit, rather than cost savings, to encourage practice nurses to participate.

## ETHICS AND DISSEMINATION

The study has been given a favourable opinion by the Cambridge Central Research Ethics Committee (REC reference number 17/EE/0136) and has been granted approval by the Health Research Authority. It is registered with the ISRCTN registry (trial ID: ISRCTN20669629, assignment date 20 March 2018, trial start date 1 April 2016).

The study involves minimal risks to patients; however, patients may be vulnerable due to age or frailty and the nature of COPD symptoms, including breathlessness and fatigue. Participants will be able to take breaks and terminate an activity if they are unable to continue. We will enquire about special support needs and participants can bring a friend or carer to any activity. Experienced researchers with appropriate training will undertake interviews and focus groups.

### Dissemination plan

We aim for timely dissemination, parallel to formal evaluation and clarifying how the outcomes will be of practical benefit to patients. Audiences include: patients and carers/families; HCPs who refer to PR and general practice managers; PR providers and service managers; PR commissioners; national interest groups, for example, British Thoracic Society; NHS England and NHS Improvement.

We will use a range of channels, working closely with the British Lung Foundation to use patient networks, publications, online information, service development and HCP engagement. We will use routes that are accessible to patient audiences, for example, Breathe Easy patient newsletters, INVOLVE website, posters in GP waiting rooms and PR classes, and village newsletters. PPI volunteers will disseminate to patient groups and networks.

Methods will include: presentations and written information for HCP forums and professional networks; presentations at regional, national and international academic and professional conferences; open access academic papers in peer-reviewed journals; web and social media updates; written feedback to participants; broadcast and print media, for example, radio and newspapers, may be considered if appropriate.

### Data deposition and curation

The study complies with the General Data Protection Regulation (2018). Data will be anonymised and entered onto secure NHS computers and University of Cambridge computers for data analysis. Signed paper consent forms containing participants' names will be stored in a locked filing cabinet on NHS premises. Only the study team will have access to participants' data. Electronic files will be password protected. Data sharing and storage will meet the requirements of the National Institutes of Health Research. Data will be securely stored in the University of Cambridge Research Repository.

**Author affiliations**
[1]Centre for Self Management Support, Cambridge University Hospitals NHS Foundation Trust, Cambridge, UK
[2]Centre for Health Services Studies, University of Kent, Canterbury, UK
[3]Department of Public Health and Primary Care, School of Clinical Medicine, University of Cambridge, Cambridge, UK
[4]Department of Engineering, University of Cambridge, Cambridge, UK
[5]COPD Team, NHS North East Essex Clinical Commissioning Group, Colchester, UK
[6]Provide Community Interest Company, Colchester, UK
[7]Cardiac/Pulmonary Rehabilitation Glenfield Hospital, University Hospitals of Leicester NHS Trust, Leicester, UK

**Acknowledgements** We thank the patients and their families who contributed to the PPI work that supported the development of this study. Thanks to Jenna Stockwell from the British Lung Foundation who is supporting the study and its implementation. Delivery of this work was supported by the Cambridge Biomedical Research Centre.

**Contributors** JF led the research team, supervised the development of the protocol, contributed to the design of the study and development of the protocol and revised the draft manuscript. He is a guarantor for the work. FE contributed to the design of the study and development of the protocol, and drafted and revised the manuscript. PW contributed to the design of the study and development of the protocol, proposed the theoretical basis for the design and supervised development of the associated methodology, and revised the draft manuscript. CD contributed to the design of the study and development of the protocol and revised the draft manuscript. She is a guarantor for the work. IW and HH contributed to the design of the study and development of the protocol and revised the draft manuscript. TD contributed to the design of the study and development of the protocol and proposed the toolkit development methodology. JW contributed to the design of the study and development of the protocol, proposed the toolkit development methodology and revised the draft manuscript. LJ, RB, SJS, JBe and LK contributed to the design of the study and development of the protocol. JBr provided technical advice regarding building and using the platform for developing the toolkit and revised the draft manuscript.

**Funding** This paper presents independent research funded by the National Institute for Health Research (NIHR) under its Research for Patient Benefit (RfPB) Programme (grant reference number PB-PG-1215-20034).

**Disclaimer** The views expressed are those of the authors and not necessarily those of the NHS, the NIHR or the Department of Health.

**Competing interests** JW reports grants from the National Institute for Health Research during the conduct of the study.

**Patient consent for publication** Not required.

**Ethics approval** Cambridge Central Research Ethics Committee.

**Provenance and peer review** Not commissioned; externally peer reviewed.

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
