## [Reviewer comments · BMJ Open]

ARTICLE DETAILS

TITLE (PROVISIONAL)	Developing an intervention to increase REferral and uptake TO pulmonary REhabilitation in primary care in patients with Chronic Obstructive Pulmonary Disease (the REsTORE study): mixed methods study protocol
AUTHORS	Early, Frances; Wilson, Patricia; Deaton, Christi; Wellwood, Ian; Dickerson, Terry; Ward, James; Jongepier, Lianne; Barlow, Ruth; Singh, Sally; Benson, John; Brimicombe, James; Kim, Lois; Haque, Hena; Fuld, Jonathan

VERSION 1 – REVIEW

REVIEWER	Tania Janaudis-Ferreira McGill University, Canada
REVIEW RETURNED	02-Jul-2018

GENERAL COMMENTS	Thank you for the opportunity to review this manuscript/protocol. This will be an important and novel study, however, the authors need to provide more details about the methods and the toolkit itself. I have some questions and suggestions for their consideration. In the introduction, the authors need to better explain why Phase 1 and 2 are needed- I suppose they would like to identify the specific barriers related to the sites in the UK but if this is the case, this information needs to be stated. Also, how do the authors expect the barriers to be different from those that have already been found in the literature? The following reference might be useful for the authors: Milner SC et al. Rate of, and barriers and enablers to, pulmonary rehabilitation referral in COPD: A systematic scoping review. Respiratory Medicine. 2018 ; 137 :103-114 In the introduction, it would be useful if the authors could expand the following sentence: “Interventions to improve referral and uptake have been tested and some positive results reported” to give the readers an idea of what these interventions involve. Also, positive benefits in terms of what? In addition, uptake and referral are two different concepts. Also, barriers related to referral are at the HCPs’ level and barriers related to Uptake are at the patients’ level. So it would be helpful to describe these interventions and their effects separately. Overall there is lack of detail related to the focus groups (how many?), sampling, analysis, qualitative approach. What is the rationale for conducting focus groups and individual interviews? Will the team ensure equal numbers of men and women (gender)
--

	in the focus groups and interviews? The barriers might be different. The authors mentioned “The toolkit will be simple and practical with information about ways to impact referral and uptake (examples may include key messages, electronic prompts/templates, links to guidelines and decision support tools and printing options).” Key messages and Templates/prompts about what? Will the toolkit be for HCPs or patients? Or both? With the information above, I am not sure how the toolkit can be helpful for patients. I think more detail on what the toolkit will include is needed. Will it be a source of resources? What type of guidelines and support tools will the authors include in the toolkit? Will the toolkit be printed or will it be available online? I understand that the project is about to develop the toolkit but the authors must have an idea of what type of toolkit they would like to develop. What are the most important things that such a toolkit should contain or do in order to be acceptable and useful for HCPs and patients? More explanation is needed about how the toolkit can improve uptake to PR (from the perspective of the patient). This link is not very clear. The authors stated that they will assess patient experience. How and about what specifically? Are they using a specific questionnaire? How will acceptability and usability be tested? More detail needs to be added in the methods. Minor comments:  - The use of the “Burden of Treatment theory” is not mentioned in the abstract. - correct word “experiences” page 5, line 22
--	--

REVIEWER	Narelle Cox La Trobe University, Australia
REVIEW RETURNED	09-Jul-2018

GENERAL COMMENTS	The authors describe in detail the protocol for a mixed methods study to develop an intervention that supports referral to and uptake of pulmonary rehabilitation. Given this study is well underway, I have just a couple of comments/questions for the authors to consider regarding the manuscript:  - Can the authors clarify why the toolkit is only intended to be trialled with practice nurses? What about other healthcare professionals? (this may relate to comment below regarding healthcare context). - Is it possible to reduce the number of abbreviations used? This would improve the reader experience (particularly where abbreviations are used only once). Also, there are a couple of abbreviations not defined eg. PPI. - It may be helpful to specify in the abstract the health care context ie. Primary care in the UK. Given primary care may not be the key
---

	driver of referrals to PR in other healthcare contexts/systems or countries. Similarly, with reference to the 'commissioners' who are taking part in the interviews; can the role/status of this professional group be clarified for readers outside the UK context. - It would be helpful if the authors could specify the date of trial registration relative to commencement of the trial. - Phase I – final paragraph: is there a word missing in the first line, prior to the word Framework? - Page 7: 'participants who declined referral and not referred' – change and to or. - The latter sections of the protocol seem to contain language more in keeping with a grant application. Might it be worth amending this? E.g. is favourability of the project from the ethics committee relevant? Use of co-applicant terminology with reference to relationship with BLF. - Supplement 6 questions only focus on negative barriers – is there a place for gaining insight into facilitators (real or perceived)?
--	---

VERSION 1 – AUTHOR RESPONSE

Reviewers' Comments to Author:

Reviewer: 1

Reviewer Name: Tania Janaudis-Ferreira

Institution and Country: McGill University, Canada Competing Interests: none

Thank you for the opportunity to review this manuscript/protocol. This will be an important and novel study, however, the authors need to provide more details about the methods and the toolkit itself. I have some questions and suggestions for their consideration.

We would like to thank both reviewers for all of their comments. We believe that in responding to these we have been able to improve the quality of our manuscript. Our specific responses to each point are below.

- In the introduction, the authors need to better explain why Phase 1 and 2 are needed- I suppose they would like to identify the specific barriers related to the sites in the UK but if this is the case, this information needs to be stated.

We have clarified the reasons for this in the Methods section under the heading Study Overview on page 6. We included it in this section rather than the Introduction because it refers to Normalisation Process Theory which, by that point, has already been described on page 5. The two reasons for Phase 1 and Phase 2 are: 1) to collect and analyse data through the lens of NPT so that we can understand how PR referral is embedded and sustained in practice; 2) to compare and contrast the perspectives and needs of 4 participant groups (patients, HCPs, PR providers and commissioners).

- Also, how do the authors expect the barriers to be different from those that have already been found in the literature?

Realistically, we may not expect to uncover a very different set of barriers, but we wish to apply the theoretical lens and the differing perspectives to those barriers in order to inform the toolkit design.

- The following reference might be useful for the authors: Milner SC et al. Rate of, and barriers and enablers to, pulmonary rehabilitation referral in COPD: A systematic scoping review. *Respiratory Medicine*. 2018; 137:103-114.

Thank you for noting this reference which we have now referred to in the Introduction on page 4.

- In the introduction, it would be useful if the authors could expand the following sentence: “Interventions to improve referral and uptake have been tested and some positive results reported” to give the readers an idea of what these interventions involve. Also, positive benefits in terms of what? In addition, uptake and referral are two different concepts. Also, barriers related to referral are at the HCPs’ level and barriers related to Uptake are at the patients’ level. So it would be helpful to describe these interventions and their effects separately.

We have clarified this on page 4 and have referred to a systematic review of these studies that we conducted which is now in press. We have made a distinction between the results from studies that measured referral and those that measured outcomes.

- Overall there is lack of detail related to the focus groups (how many?), sampling, analysis, qualitative approach.

We have clarified the number of focus groups and focus group participants on page 8. The sampling strategy is driven initially by the utilisation rates of PR services as described on page 7 and then convenience sampling is used within those areas. We felt it was important and necessary to take a pragmatic approach to the sampling as we anticipated that securing the participation of some groups, particularly patients who have declined PR and clinicians in primary care with busy and demanding schedules, could be challenging and we wanted to ensure maximum flexibility for recruitment within our targeted population. We have amended the description of the sampling strategy in order to clarify this on page 8. The methodology for conducting the qualitative analysis follows the Framework approach as described by Ritchie et al and we have referenced this on page 9.

- What is the rationale for conducting focus groups and individual interviews?

This was a pragmatic decision. We believe in the benefits of focus groups for generating a variety of viewpoints through interaction but believed that it would probably not be practicable to engage sufficient numbers patients who had already declined PR or not been referred to PR in one place at one time, so for practical reasons we planned interviews with this group. For HCPs we allowed for focus groups where possible in order to draw out shared experiences among this professional group; however we expected the practicalities of this to be challenging and so allowed for interviews also. We have clarified this on page 8.

- Will the team ensure equal numbers of men and women (gender) in the focus groups and interviews? The barriers might be different.

We agree that different barriers may apply and will aim to achieve a gender balance. We have clarified this on page 7.

- The authors mentioned “The toolkit will be simple and practical with information about ways to impact referral and uptake (examples may include key messages, electronic prompts/templates, links to guidelines and decision support tools and printing options).” Key messages and Templates/prompts about what?

We have given examples of key messages about the benefits of PR and prompts for PR referral on page 10.

- Will the toolkit be for HCPs or patients? Or both? With the information above, I am not sure how the toolkit can be helpful for patients. I think more detail on what the toolkit will include is needed. Will it be a source of resources? What type of guidelines and support tools will the authors include in the toolkit? Will the toolkit be printed or will it be available online? I understand that the project is about to develop the toolkit but the authors must have an idea of what type of toolkit they would like to develop. What are the most important things that such a toolkit should contain or do in order to be acceptable and useful for HCPs and patients? More explanation is needed about how the toolkit can improve uptake to PR (from the perspective of the patient). This link is not very clear.

We have expanded on this and indicated more clearly how the toolkit will be used and the link with uptake on page 10. The toolkit is intended for direct use by healthcare professionals in primary care with the aim of enhancing their understanding of PR, providing resources that support a meaningful and motivational conversation about PR with the patient and facilitating the referral process. To this end we envisage that it will include materials and information for patients that would be shared with them by the HCP, rather than the patient accessing the toolkit directly. We do have ideas about what we think the likely content will be based on the literature and on our own experience of talking to patients and clinicians about PR. It is however very important that we don't pre-empt the design process, both in terms of content and medium, which must be co-produced with users. Based on what we know from the literature about barriers to referral and uptake we would envisage important features to include: clear information about PR and its benefits and eligibility criteria, prompts and tools to facilitate the referral process and tools to support an exploration of patient's needs and beliefs about PR and to motivate and encourage uptake.

- The authors stated that they will assess patient experience. How and about what specifically? Are they using a specific questionnaire?

We have developed a non-validated questionnaire for this purpose and have attached this as additional supplementary material (Supplement 9).

- How will acceptability and usability be tested? More detail needs to be added in the methods.

We have clarified this on pages 10-11. Acceptability and usability criteria will be developed as part of the iterative design process. The subsequent methods of testing acceptability and usability will be those best suited to measure those criteria. Therefore, at this stage, we cannot pre-determine what those methods will be. However, we envisage considering a range of approaches including retrospective methods, e.g. written feedback and interviews, and real time methods, e.g. thinking aloud techniques and objective methods, e.g. user observation. Testing will take place after each iteration of the design process

Minor comments:

- The use of the "Burden of Treatment theory" is not mentioned in the abstract.

Thank you for noting this omission. This is now included in the abstract.

- correct word "experiences" page 5, line 22.

This has now been corrected.

Reviewer: 2

Reviewer Name: Narelle Cox

Institution and Country: La Trobe University, Australia Competing Interests: None declared

The authors describe in detail the protocol for a mixed methods study to develop an intervention that supports referral to and uptake of pulmonary rehabilitation.

Given this study is well underway, I have just a couple of comments/questions for the authors to consider regarding the manuscript:

- Can the authors clarify why the toolkit is only intended to be trialled with practice nurses? What about other healthcare professionals? (this may relate to comment below regarding healthcare context).
It is our experience that practice nurses are the main referrers to PR in primary care in the UK. However, on reflection it is right not to assume that this will always be the case. We have therefore changed 'nurse' to 'HCP' on page 11 so that we can identify the most suitable clinician to test the toolkit in each practice.

- Is it possible to reduce the number of abbreviations used? This would improve the reader experience (particularly where abbreviations are used only once). Also, there are a couple of abbreviations not defined e.g. PPI.

We have reviewed and amended the use of abbreviations throughout the manuscript.

- It may be helpful to specify in the abstract the health care context ie. Primary care in the UK. Given primary care may not be the key driver of referrals to PR in other healthcare contexts/systems or countries.

We have noted this in the abstract.

- Similarly, with reference to the 'commissioners' who are taking part in the interviews; can the role/status of this professional group be clarified for readers outside the UK context.

We clarified this on page 7.

- It would be helpful if the authors could specify the date of trial registration relative to commencement of the trial.

Registration was applied for retrospectively on 29/1/18 and the study was assigned on 20/3/18. The registered trial start date is 1/4/2016. We have noted this in the abstract and in the Ethics and Dissemination section on page 11. Recruitment to the study opened on 1/6/17 as indicated in the trial registration details.

- Phase I – final paragraph: is there a word missing in the first line, prior to the word Framework?

Thank you for noting this. In fact there is no word missing as 'framework' is the name for the process we are following. We have added quotation marks to the word 'framework' in order to more clearly indicate that this is the term used for the process.

- Page 7: 'participants who declined referral and not referred' – change and to or.

We have made this change.

- The latter sections of the protocol seem to contain language more in keeping with a grant application. Might it be worth amending this? E.g. is favourability of the project from the ethics committee relevant? Use of co-applicant terminology with reference to relationship with BLF.

We were guided by the journal template for protocol manuscripts but do understand this comment. We have deleted the word 'co-applicant' on page 12 to describe the British Lung Foundation.

- Supplement 6 questions only focus on negative barriers – is there a place for gaining insight into facilitators (real or perceived)?

We agree that the wording of question 7 relates to negative barriers, although the prompts below this do target facilitators as well. In order to distinguish clearly between the two we have amended this schedule so that there is a new question 8 which refers independently to facilitators. We have also made a corresponding change to supplement 5 where the same issue arises.

VERSION 2 – REVIEW

REVIEWER	Tania Janaudis-Ferreira, Assistant Professor McGill University, Canada
REVIEW RETURNED	05-Sep-2018

GENERAL COMMENTS	Thank you again for the opportunity to review this protocol. The authors have addressed all my comments and suggestions.
--

REVIEWER	Narelle Cox La Trobe University, Melbourne, Australia
REVIEW RETURNED	12-Sep-2018

GENERAL COMMENTS	The authors have provided additional content regarding the methods that add to the clarity and detail of the protocol. I have no additional queries.
--